# Fostering participation motivation and multidisciplinary teamwork collaboration through hospital culture and team leadership in Chinese tertiary public hospitals—A cross-sectional study

Zhengnan Meng[1], Jiamin Tang[1], Jie Yang[1], Huaineng Wu[2], Linlin Cui[3], Simin Zhu[2], Xiaohe Wang[1]*, Xianhong Huang[1]*

1 School of Public Administration, Hangzhou Normal University, Hangzhou, China, 2 Affiliated Mental Health Center and Hangzhou Seventh People's Hospital, Zhejiang University School of Medicine, Hangzhou, China, 3 Hangzhou International Urbanology Research Center and Center for Urban Governance Studies, Hangzhou, China

* hxh974291@163.com (XH); xhewang@163.com (XW)

## Abstract

Many modern diseases require more than one discipline for effective treatment, making multidisciplinary teamwork (MDT) essential. However, research on MDT in tertiary public hospitals in China is limited. These large healthcare institutions require effective collaboration among various departments to manage complex cases. Therefore, this study examined the effects of hospital culture, team leadership, and participation motivation on the MDT behavior of healthcare professionals to enhance MDT and improve related services. We conducted a questionnaire survey of 425 multidisciplinary team members in tertiary public hospitals in Hangzhou. T-tests, analysis of variance, and hierarchical linear regression were used to analyze the state of healthcare professionals' MDT behaviors and the associated factors. A path analysis using a Structural Equation Model was used to explore and verify the effects of hospital culture, team leadership, and participation motivation on MDT behavior, as well as their underlying mechanisms. The findings revealed significant positive effects of hospital culture, team leadership, and participation motivation on MDT behavior. An SEM path analysis confirmed that these factors directly influence MDT behavior. Moreover, hospital culture and team leadership indirectly affected MDT behavior through participation motivation. This study demonstrated that a positive hospital culture and team leadership significantly enhanced MDT, with participation motivation mediating this relationship. These findings suggest that hospital leadership should promote a proactive and harmonious hospital culture and facilitate the development of team leaders' management skills. Furthermore, exploring diverse incentives to increase healthcare professionals' motivation for participation is essential for advancing MDT.

**Data availability statement:** The datasets generated and analyzed during the current study contain potentially identifying information. Although direct identifiers (e.g., names, addresses) have been removed, the combination of narrow age bands, gender, professional title, and department codes from a limited number of tertiary hospitals may still allow for participant re-identification. As a result, the data cannot be made publicly available. However, the data can be made available to qualified researchers who submit a formal request. To request access to the data, interested researchers should: (1) Submit a written request to the Research Ethics Committee of Hangzhou Normal University at kejichu@hznu.edu.cn, clearly stating the purpose of their research and how the data will be used. (2) Include a brief description of the research plan, specifying the type of data required. (3) Agree to the terms and conditions set by the Ethics Committee for ensuring data privacy and security. The Research Ethics Committee will evaluate each request based on ethical guidelines, the purpose of the research, and the potential risks of re-identification. If the request is approved, access to the data will be granted under strict confidentiality and security measures.

**Funding:** This research was funded by the National Natural Science Foundation of China Project (grant numbers 72274051 and 71974050) and Hangzhou Medicine and Health Science and Technology Program(grant number ZD20220099). The funders had no role in study design, data collection and analysis, decision to publish, or preparation of the manuscript.

**Competing interests:** The authors declare no competing interests.

## Introduction

The medical treatment model is continuously evolving. Traditionally, the diagnosis and treatment of diseases have depended on a single-discipline approach. However, recent studies have shown that this method fails to address patients' comprehensive needs [1,2]. As society advances, the medical model has transitioned to a bio-psycho-social approach, prompting healthcare professionals to consider the whole patient and address complex issues by integrating medical knowledge [3]. To enhance the quality of care, healthcare institutions are increasingly adopting multidisciplinary teams (MDTs). An MDT is a stable group of at least three clinicians from different specialties who independently assess the patient and make discipline-specific decisions, sharing information through regular meetings and emphasizing professional boundaries and standardized processes in a parallel collaborative model [4] This allows healthcare professionals to analyze issues from multiple perspectives and deliver effective patient care. Studies indicate that MDTs can increase patient survival [5], enhance satisfaction with medical services [6], and improve clinicians' diagnostic skills [7] and job satisfaction [8]. However, challenges, such as inadequate inter-departmental teamwork [9] and low engagement of healthcare professionals in MDT [10], persist, preventing the full realization of the MDT model in medical practice. In China, approximately 70% of adverse events result from teamwork failures [11], significantly wasting healthcare resources [12]. Therefore, identifying the factors that influence MDT in tertiary public hospitals and their underlying mechanisms is essential to promote the use of MDT in clinical practice, enhancing healthcare quality, and facilitating the development of disease treatment development in public hospitals.

Research on factors influencing MDT has largely concentrated on internal organizational elements, such as hospital culture, team leadership, participation motivation, and organizational climate [13–15]. However, existing studies have notable gaps. First, few studies have examined the cooperative behaviors of MDT healthcare professionals as an outcome variable, although understanding these behaviors is essential because healthcare professionals are the pillar of MDT. Second, although hospital culture, team leadership, and participation motivation are critical factors impacting MDT, comprehensive research on their pathways of influence remains insufficient. Furthermore, hospital culture and team leadership influence motivation to participate [16,17]. However, studies examining the mediating role of participation motivation in the effects of hospital culture and team leadership on MDT are lacking. Therefore, this study investigated the impact of hospital culture, team leadership, and participation motivation on MDT behavior.

To address the limitations of previous studies, this study adopted the Input-Process-Output (IPO) model as a theoretical framework to analyze the influencing paths affecting MDT. The IPO model, a well-established framework for examining team collaboration, states that team inputs directly impact outputs and indirectly influence team processes [18]. Inputs encompass various factors that influence team members, such as individual characteristics (e.g., personality and skills), team dynamics (e.g., structure and cohesion), and external conditions (e.g., organizational climate and support). Processes pertain to the behaviors team members engage in to achieve their goals, whereas outputs reflect team

performance and individual emotional responses, such as satisfaction and commitment [19]. This model has frequently been used in healthcare research to examine the direct effects of input factors, such as knowledge, skills, and organizational support, on output factors, such as team member satisfaction and effectiveness, as well as their indirect effects through process factors such as interpersonal relationships, communication skills, and team reflexivity [20,21]. MDT behaviors in tertiary public hospitals are shaped by input factors, such as hospital culture, team leadership, and organizational structure, as well as process factors, including participation motivation and interaction intention among team members.

This study focused on healthcare professionals engaged in MDT in tertiary public hospitals in Hangzhou, China. This study used the IPO model to analyze the effects of hospital culture, team leadership, and participation motivation on MDT behavior, including the mediating role of participation motivation. This empirical study aimed to elucidate the complex relationships between input and output factors and develop a model outlining factors influencing MDT behavior. A structural equation analysis was used to overcome the limitations of previous research regarding the factors and pathways affecting MDT behavior. This study sought to improve the quality of medical diagnosis and treatment and provide a theoretical framework for policymakers to guide the development of MDT in tertiary public hospitals.

## Literature review and hypothesis development

### The influence of hospital culture on MDT behavior

Hospital culture encompasses values, beliefs, and attitudes that connect healthcare professionals [22]. Studies have shown that hospital culture positively influences teamwork. A longitudinal study in Norwegian surgical wards revealed that collaborative cultural environments fostered trust among team members, promoting participation in teamwork [23]. Qin et al. [24] examined the missions, visions, and values of five leading hospitals and found that a patient-centered culture significantly enhanced collaboration among healthcare professionals. Moreover, a hospital culture that prioritizes employee satisfaction, team safety [25], effective communication and mutual collaboration [26], and innovation significantly enhances teamwork among healthcare professionals across different departments [27]. Thus, existing research suggests that the core values and ethical standards of hospital culture positively influence the MDT behaviors of healthcare professionals by fostering a supportive work environment. Therefore, we proposed the following hypothesis:

H1: Hospital culture positively affects MDT behavior.

### The influence of team leadership on MDT behavior

Team leadership encompasses a leader's ability to effectively communicate the team's mission and objectives while guiding and influencing members toward achieving these goals [28]. Research shows that the leadership skills of the individual overseeing MDT are crucial for fostering cooperation among team members [29]. Beiboer et al. [30] found that strong team leadership enhanced constructive collaboration in clinical settings. Akinsola et al. [31] demonstrated that effective team leadership facilitates communication among individuals, resolves issues, and promotes teamwork. Moreover, a study examining teamwork among oncology nursing teams identified team leadership as a key factor influencing nurses' teamwork behavior [32]. Tremblay et al. [33] evaluated the collaborative conditions in oncology MDT and revealed that professional attributes, resource allocation, policy support, and team leadership were vital for effective team functioning. This suggests that the leadership exhibited by team leaders in areas such as goal setting and conflict resolution can positively impact MDT among healthcare professionals. Therefore, we proposed the following hypothesis:

H2: Team leadership positively affects MDT behavior.

### The influence and mediating roles of participation motivation

Participation motivation is an internal drive fueled by the desire to achieve specific goals [34]. The motivation of healthcare professionals to participate is essential for delivering high-quality healthcare services and fostering effective teamwork

[35]. Yahang et al. examined how participation motivation indirectly improves team performance by influencing teamwork behavior [36]. The motivation of healthcare professionals to engage in their work significantly impacts their MDT behavior. Research [37] indicates that the integrating and reforming of the healthcare service system can boost healthcare professionals' motivation, thereby enhancing their enthusiasm for teamwork.

Furthermore, participation motivation is a key factor linking hospital culture and MDT behavior. Chmielewska et al. [38] discovered that a hospital culture characterized by harmonious interaction enhanced healthcare professionals' organizational commitment and participation motivation to participate, thereby improving overall hospital performance. Huang et al. [39] analyzed the effects of hospital culture on patient-centered care. They reported that effective internal communication and a culture of innovation significantly boosted healthcare professionals' motivation to engage in their work, thereby facilitating cross-team collaboration.

Moreover, studies have shown that participation motivation is crucial in linking team leadership and MDT behavior. A survey conducted in various regions of Uganda revealed that leadership styles were significantly associated with job satisfaction, motivation for participation and teamwork among healthcare professionals [40]. Subramaniam et al. [41] demonstrated that participation motivation mediated the relationship between leadership and safety behavior among nurses. Chaman et al. [42] found that employees' internal motivation for participation mediated the relationship between team leadership style and knowledge sharing, which was essential for fostering organizational cooperation. Tran and Hoang [43] discovered that transformational leadership positively influenced job performance, with employee participation serving as a mediating factor.

These findings indicate that hospital culture, team leadership, participation motivation, and MDT behavior are interrelated. Furthermore, participation motivation mediates the relationship between various antecedent variables, such as team leadership, hospital culture, social support, and professional resources, and outcome variables, such as cooperative behavior, job satisfaction, turnover intention, and job performance. Therefore, we proposed the following hypotheses:

H3: Participation motivation positively affects MDT behavior.

H4a: Participation motivation mediates the relationship between hospital culture and MDT behavior.

H4b: Participation motivation mediates the relationship between team leadership and MDT behavior.

The theoretical model of this study is presented in Fig 1.

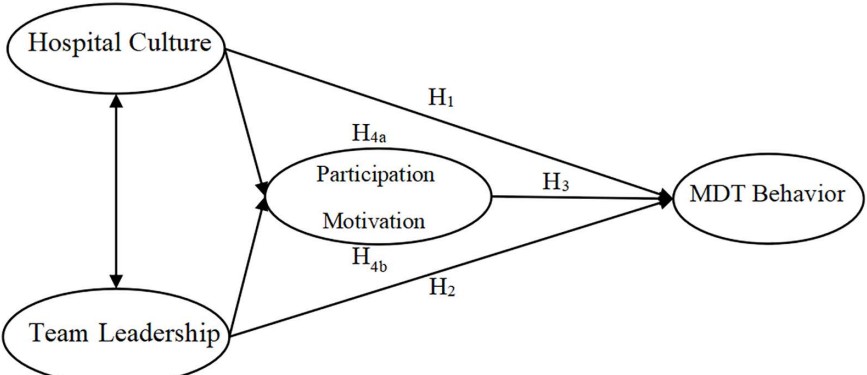

**Fig 1. Theoretical model of the relationships between hospital culture, team leadership, and MDT behavior of healthcare professionals in tertiary public hospitals.**

## Methods

### Participants and procedures

With its rich medical resources, Hangzhou has become a leading city in implementing MDT models in tertiary hospitals. Therefore, this study focused on tertiary public hospitals in Hangzhou and used purposive sampling to survey doctors, nurses, and medical technicians engaged in MDT collaboration.

Participants were chosen from tertiary public hospitals in Hangzhou that have adopted an MDT model for diagnosis and treatment. The inclusion criteria were (1) being a clinical doctor, nurse, or medical technician engaged in MDT and (2) being currently employed and having participated in MDT for more than six months during the survey. The exclusion criteria were (1) being a healthcare professional visiting for training or education; (2) being a healthcare professional currently off duty; (3) being a healthcare professional who only attend MDT case discussions as listener and does not have any experience participating in MDT; and (4) being administrative or support personnel involved in MDT organization and management.

Regarding sample-size calculation we followed Thompson's SEM guideline: the ratio of observed variables to respondents should be at least 10:1, with 15:1–20:1 considered optimal [44]. The present study comprises 16 observed variables assuming a conservative figure of 20 cases per variable yielded 320 respondents. Anticipating incomplete or inaccurate responses, we inflated this figure by an additional 20% to 384. From January 10, 2023 to July 20, 2023, we distributed 500 questionnaires and obtained 452 valid responses, corresponding to an effective response rate of 90.4%. To ensure adequate statistical power, a post hoc power analysis was conducted using G*Power 3.1 to assess the adequacy of the achieved sample size (6 predictors, $f^2 = 0.15$, $α = 0.05$, $N = 452$) [45]. The achieved statistical power $(1 − β)$ was 0.99. This finding indicates that the study had more than sufficient power to detect medium-sized effects, thereby supporting the robustness of the statistical inference [46].

### Measures

A general survey form was used to collect essential demographic and occupational information from healthcare professionals involved in MDT. The form included six items: gender (men, women), age (≤ 30 years, 31–40 years, 41–50 years, 51–60 years, ≥ 61 years), professional category (clinical doctor, nurse, medical technician), professional title (intermediate, associate senior, full senior), department (internal medicine, surgery, obstetrics and gynecology, pediatrics, medical technology, rehabilitation, other), and type of MDT participation (outpatient, inpatient, both, other).

Hospital culture was measured using a four-item unidimensional scale adapted from The Hospital Culture Questionnaire developed by Körner et al. [47]. The items cover departmental collaboration, information sharing, academic exchange, and social responsibility (e.g., "Different departments within the hospital often work together to implement innovative changes"). Responses were rated on a five-point Likert scale (1 = highly inconsistent; 5 = highly consistent), with higher scores indicating a more favorable hospital culture. The composite indices were calculated as the simple arithmetic mean of the item scores (sum of item scores/ number of items). The scale demonstrated strong reliability in this study, with a Cronbach's alpha coefficient of 0.829, a composite reliability (CR) value of 0.81, factor loadings ranging from 0.56 to 0.84, and an average variance extracted (AVE) value of 0.52.

Leadership was measured using a four-item unidimensional scale adapted from "A Tumor Leadership Assessment instrument" (ATLAS) developed by Jalil et al. [48]. The items focus on the leader's abilities in "coordinating and executing tasks" "promoting team member interaction" and "managing conflict". A sample item is "The team leader ensures that all cases requiring discussion are reviewed promptly." Responses were rated on a five-point Likert scale (1 = highly inconsistent; 5 = highly consistent), with higher scores indicating stronger team leadership. The composite indices were calculated as the simple arithmetic mean of the item scores (sum of item scores/ number of items). The scale demonstrated strong reliability and validity in this study, with a Cronbach's alpha coefficient of 0.913, a CR value of 0.91, factor loadings between 0.80 and 0.87, and an AVE value of 0.73.

Participation motivation was measured using a four-item unidimensional scale adapted from The Work Extrinsic and Intrinsic Motivation Scale (WEIMS) developed by Tremblay et al. [49]. The items encompass both intrinsic motives—such as the desire to enhance one's skills, improve communication and collaboration, and pursue continuous learning and growth—and extrinsic motives, such as economic benefits. A sample item is "Participating in MDT can elevate my clinical expertise." Responses were rated on a five-point Likert scale (1 = highly inconsistent; 5 = highly consistent), with higher scores indicating higher participation motivation. The composite indices were calculated as the simple arithmetic mean of the item scores (sum of item scores/ number of items). The scale demonstrated good reliability and validity in this study, with a Cronbach's alpha coefficient of 0.739, a CR value of 0.84, factor loadings ranging from 0.72 to 0.85, and an AVE value of 0.64.

MDT behavior was measured using a five-item simplified version of the Interprofessional Teamwork Scale, which was translated and adapted by Jin et al. [50]. The items address shared decision-making, communication and collaboration, experience sharing, conflict resolution, and continuous improvement. A sample item is "MDT members participate equally in the decision-making process for treatment plans." Responses were rated on a five-point Likert scale (1 = highly inconsistent; 5 = highly consistent), with higher scores indicating more efficient MDT behavior. The composite indices were calculated as the simple arithmetic mean of the item scores (sum of item scores/ number of items). The scale demonstrated strong reliability and validity in this study, with a Cronbach's alpha coefficient of 0.882, a CR value of 0.89, factor loadings between 0.72 and 0.85, and an AVE value of 0.61.

The full questionnaire is provided in the S1 Appendix, as Supporting Information.

## Ethical considerations

This study was reviewed and approved by the Research Ethics Committee of Hangzhou Normal University on March 3, 2022, including the study protocol, informed consent form, and questionnaire. The research adhered to the ethical guidelines outlined in the 1964 Declaration of Helsinki and its subsequent amendments. Prior to data collection, all participants were verbally informed about the study objectives, procedures, confidentiality protections, voluntary participation, and their right to withdraw at any time without negative consequences. Verbal informed consent was obtained in the presence of a trained investigator, who served as a witness and immediately documented the consent in a standardized electronic log system. The log included the participant's initials, date, and time of consent. All records were securely stored in an encrypted database accessible only to the research team, ensuring confidentiality and traceability. This verbal consent procedure, along with the study protocol and questionnaire, was reviewed and formally approved by the Ethics Committee of Hangzhou Normal University (Approval No. 2022−1121) prior to the initiation of the study.

## Bias control

To minimize the impact of potential biases on the results, this study implemented comprehensive control measures throughout the entire data collection and analysis process. During the sampling phase, purposive sampling was used to cover multiple tertiary public hospitals in Hangzhou. Strict adherence to the inclusion criteria was maintained, while ensuring representativeness across different departments (internal medicine, surgery, medical technology, etc.) and professional categories (doctors, nurses, medical technicians) to reduce selection bias. During the data collection phase, the questionnaire was revised after pre-surveys to ensure the clarity and understandability of its items. Investigators conducted one-on-one surveys after unified training, using standardized instructions to reduce comprehension biases. Anonymous response methods were adopted to encourage truthful feedback from respondents, thereby reducing reporting bias. In statistical analysis, demographic and occupational characteristic variables identified as statistically significant through univariate analysis (e.g., professional title, MDT participation type) were included as covariates in hierarchical linear regression and structural equation models to control for the interference of confounding bias. Additionally, scales used in the study, such as those for hospital culture and team leadership, all passed reliability and validity tests (Cronbach's α

coefficient > 0.7, factor loadings > 0.5), ensuring the stability and effectiveness of measurement tools and reducing measurement bias.

## Statistical analysis

The data analysis was performed using SPSS 27.0 and Amos 25.0. The reliability of the questionnaire was evaluated using CR values and internal consistency (Cronbach's alpha coefficient), whereas structural validity was assessed through a confirmatory factor analysis. Content validity was determined using the item-to-total correlation method. Descriptive statistics were used to outline the demographic and occupational characteristics of the participants, along with the current state of hospital culture and team leadership. Categorical variables were reported as frequencies and proportions, whereas continuous variables with a normal distribution were summarized using the mean and standard deviation. The proportion of missing values ranged from 1.1% to 2.9% across all study variables, indicating a low level of missingness. Little's MCAR test indicated that the data were missing completely at random (MCAR) (P > 0.05) [51]. Given this low and random pattern of missing data, we applied multiple imputation using five imputed datasets. The imputation model was based on a multivariate normal regression approach with relevant auxiliary variables [52]. Rubin's rule was used to pool estimates across datasets [53]. In addition, sensitivity analyses using complete case analysis yielded results consistent with the multiply imputed datasets, supporting the robustness of our findings. Univariate analysis was used to compare participation motivation and MDT behavior scores among MDT members with varying demographic and occupational characteristics. For normally distributed variables, t-tests were employed for comparisons between two groups, whereas a one-way analysis of variance (ANOVA) was used for comparisons involving three or more groups. A hierarchical linear regression analysis using the Enter method was applied to identify the primary factors influencing healthcare professionals' participation in MDT behavior in tertiary public hospitals.

A structural equation model was developed using Amos 25.0 to examine the direct and indirect effects of hospital culture, team leadership, and participation motivation on MDT behavior. The bootstrap method was used to test the mediating effect of participation motivation. The model comprised two primary pathways: hospital culture influencing participation motivation, which in turn affected cooperative behavior; and team leadership impacting participation motivation, which also affected cooperative behavior. The model fit was evaluated and adjusted based on modification indices.

## Results

### Participant characteristics

A total of 500 questionnaires were distributed to healthcare professionals engaged in MDTs across tertiary public hospitals in Hangzhou. Of these, 452 valid questionnaires were returned and included in the analysis, yielding a valid response rate of 90.4%. Forty-eight questionnaires were excluded due to either refusal to participate (n = 12) or incomplete/invalid responses (n = 36). The participant flow is illustrated in Fig 2. The participants' demographic characteristics are presented in Table 1.

### Correlations between hospital culture, team leadership, participation motivation, and MDT behavior

As shown in Table 2, hospital culture was positively correlated with team leadership, participation motivation, and MDT behavior. Team leadership was positively correlated with participation motivation and MDT behavior. Furthermore, participation motivation was positively correlated with MDT behavior.

### Hierarchical linear regression results for MDT behavior

First, unordered categorical data that demonstrated statistical significance in the univariate analysis of healthcare professionals' cooperative behavior, such as professional category, title, and type of MDT participation, were dummy-coded.

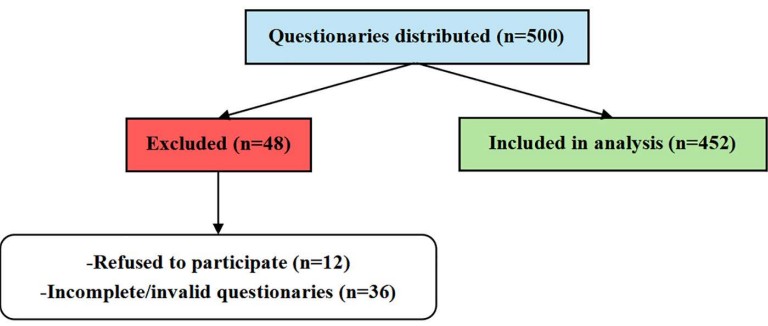

**Fig 2. Flow of participants through the study.**

**Table 1. Participant characteristics (N = 452).**

| Variable | Category | Frequency | Percentage (%) |
|---|---|---|---|
| **Gender** | Male | 246 | 54.4 |
| | Female | 206 | 45.6 |
| **Age (years)** | ≤ 30 | 15 | 3.3 |
| | 31–40 | 176 | 38.9 |
| | 41–50 | 193 | 42.7 |
| | ≥ 51 | 68 | 15.1 |
| **Department** | Internal medicine | 168 | 37.2 |
| | Surgery | 115 | 25.4 |
| | Obstetrics and gynecology | 15 | 3.3 |
| | Pediatrics | 30 | 6.6 |
| | Medical technology | 98 | 21.7 |
| **Professional Category** | Clinical doctor | 357 | 79.0 |
| | Nurses and medical technicians | 95 | 21.0 |
| **Professional title** | Intermediate | 116 | 25.7 |
| | Associate senior | 193 | 42.7 |
| | Full senior | 143 | 31.6 |
| **MDT participation type** | Outpatient | 14 | 3.1 |
| | Inpatient | 253 | 56.0 |
| | Outpatient and inpatient | 174 | 38.5 |
| | Other | 11 | 2.4 |

**Table 2. Correlations between hospital culture, team leadership, participation motivation, and MDT behavior.**

| Variable | Mean±SD | Hospital culture | Team leadership | Participation motivation | MDT behavior |
|---|---|---|---|---|---|
| **Hospital culture** | 3.74±0.62 | 1 | | | |
| **Team leadership** | 4.13±0.56 | 0.540** | 1 | | |
| **Participation motivation** | 3.80±0.56 | 0.544** | 0.541** | 1 | |
| **MDT behavior** | 3.98±0.56 | 0.490** | 0.651** | 0.554** | 1 |

** $P < 0.01$ (two-tailed).

Hierarchical linear regression was conducted with healthcare professionals' cooperative behavior as the dependent variable. The independent variables were categorized into four layers: the first layer comprised demographic variables, the second layer added hospital culture variables, the third layer added team leadership variables, and the fourth layer added participation motivation variables. The results indicated that incorporating demographic characteristics, hospital culture, team leadership, and participation motivation in the regression model produced statistically significant changes in $\Delta R^2$. Analyzing the changes in $\Delta R^2$ values revealed that hospital culture exerted a more substantial influence on healthcare professionals' MDT behavior than demographic characteristics, team leadership, and participation motivation, explaining 23.3% of the variance in this behavior.

As shown in Table 3, an analysis of the independent variables in the fourth-layer model revealed that among healthcare professionals holding the same highest title, those with an intermediate title exhibited the highest scores for cooperative behavior. Hospital culture positively influenced the MDT behavior of healthcare professionals ($\beta = 0.126$, $P = 0.004$). Moreover, team leadership significantly enhanced MDT behavior ($\beta = 0.443$, $P < 0.001$), and participation motivation further increased MDT behavior ($\beta = 0.252$, $P < 0.001$).

## Structural equation model results for MDT behavior

**Model construction.** A structural equation model was developed using hospital culture, team leadership, participation motivation, and MDT behavior as latent variables. The specific dimensions of hospital culture, team leadership, participation motivation, and MDT behavior served as observed variables (Fig 3). As shown in Table 4, the model fit results revealed a $\chi^2$/df ratio of 3.137, with all goodness-of-fit indices (GFI, NFI, CFI, TLI) exceeding 0.9. RMSEA was below 0.08, and AGFI was above 0.8, indicating a good model fit.

**Table 3. Hierarchical regression results for MDT behavior.**

| Variable | First block | Second block | Third block | Fourth block |
|---|---|---|---|---|
| | Standardized β | Standardized β | Standardized β | Standardized β |
| **Post (reference: clinical doctor)** | | | | |
| Nurse | 0.047 | 0.015 | 0.010 | 0.022 |
| Medical technician | −0.087 | −0.016 | −0.006 | −0.007 |
| **Professional title (reference: full senior)** | | | | |
| Intermediate | −0.057 | −0.095* | −0.142** | −0.142** |
| Associate senior | −0.171* | −0.195** | −0.201** | −0.197** |
| **MDT participation type (reference: outpatient and inpatient)** | | | | |
| Outpatient | −0.107* | −0.043 | −0.012 | −0.013 |
| Inpatient | −0.031 | 0.050* | 0.072 | 0.067 |
| Other | −0.125* | −0.091* | −0.026 | −0.045 |
| Hospital culture | | 0.506** | 0.217** | 0.126* |
| Team leadership | | | 0.533** | 0.443** |
| Participation Motivation | | | | 0.252** |
| $R^2$ | 0.062 | 0.295 | 0.488 | 0.527 |
| $F$ | 4.193** | 23.216** | 46.836** | 49.118** |
| $\triangle R^2$ | 0.062 | 0.233 | 0.193 | 0.193 |
| $\triangle F$ | 4.193** | 146.744** | 166.432** | 36.144** |
| VIF$_{max}$ | 1.446 | 1.452 | 1.526 | 1.739 |

*$P < 0.05$, ** $P < 0.001$. VIF$_{max}$, variance inflation factor maximum.

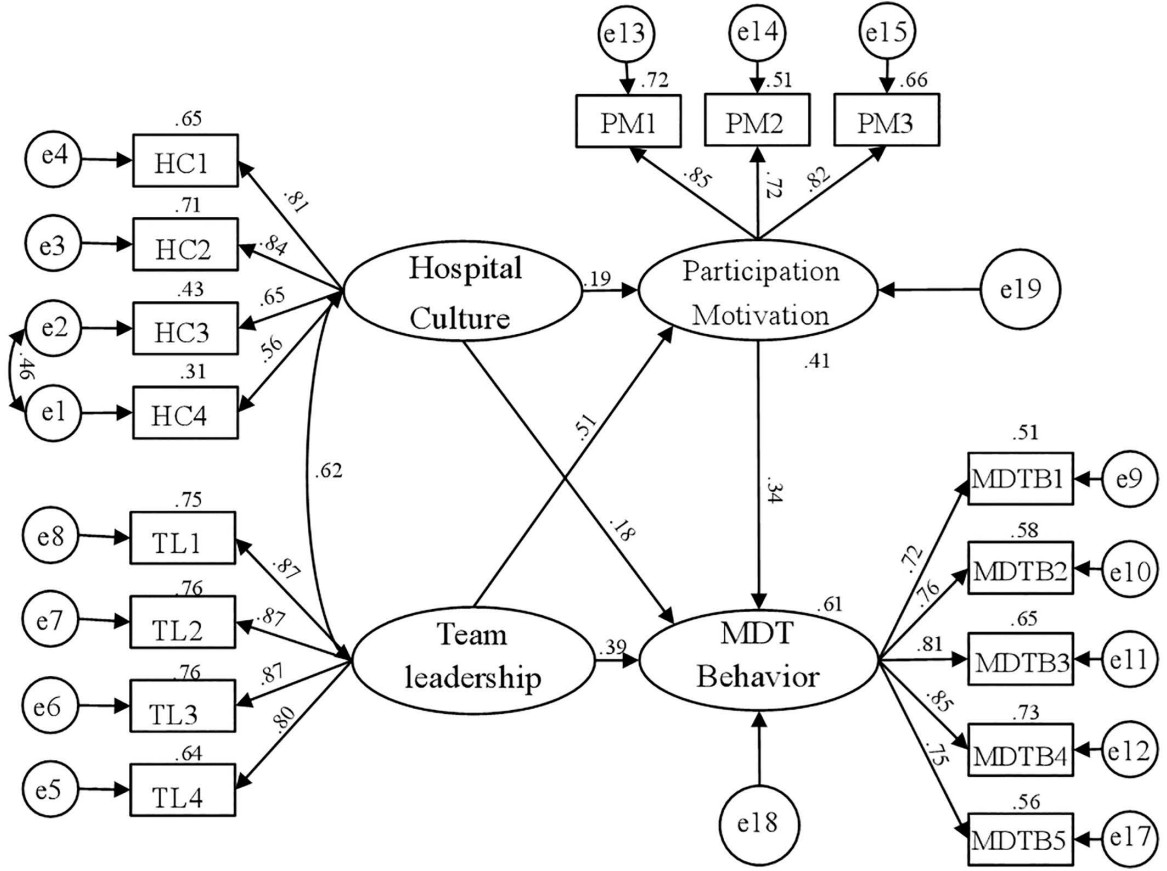

**Fig 3. Model of the influencing mechanism of MDT behavior among healthcare professionals.**

**Table 4. Structural equation model fit results.**

| Fit indices | Standards | Model fit |
|---|---|---|
| $\chi^2/df$ | 3–5 acceptable; 1–3 good | 3.137 |
| GFI | > 0.8 acceptable; > 0.9 good | 0.922 |
| AGFI | > 0.8 acceptable; > 0.9 good | 0.891 |
| RMSEA | < 0.1 acceptable; < 0.08 good | 0.069 |
| IFI | > 0.8 acceptable; > 0.9 good | 0.954 |
| CFI | > 0.8 acceptable; > 0.9 good | 0.954 |
| NFI | > 0.8 acceptable; > 0.9 good | 0.934 |
| TLI | > 0.8 acceptable; > 0.9 good | 0.943 |

**Impact pathways of MDT behavior.** As shown in Table 5, the standardized effect analysis results indicated that the culture of tertiary public hospitals and team leadership significantly enhanced MDT behavior. Furthermore, hospital culture and team leadership directly influenced MDT behavior and had indirect effects through participation motivation, supporting H1 and H2. Moreover, participation motivation significantly and directly affected MDT behavior, supporting H3.

**Mediation effects of participation motivation.** The mediation effect of participation motivation on MDT behavior among healthcare professionals was assessed using the bootstrap method, with a sample size of 2000 and a 95%

**Table 5. Influencing path coefficients on MDT behavior.**

| Relationship between variables | Standardized direct effect | Standardized indirect effect | Standardized total effect | P | Supported hypothesis |
|---|---|---|---|---|---|
| Hospital culture → Participation motivation → MDT behavior | 0.180 | 0.062 | 0.242 | P=0.001 | H1 |
| Team leadership → Participation motivation → MDT behavior | 0.393 | 0.171 | 0.564 | P<0.001 | H2 |
| Participation motivation → MDT behavior | 0.335 | / | 0.335 | P<0.001 | H3 |

confidence interval. The analysis used the maximum likelihood estimation approach. The results revealed total effect values of 0.242 for hospital culture and 0.564 for team leadership on MDT behavior. The confidence interval analysis did not include zero, indicating that the overall mediation effect was significant. Hospital culture and team leadership directly affect MDT behavior with effect values of 0.180 and 0.393, respectively, and confidence intervals not including 0, indicating significant direct effects. Additionally, hospital culture and team leadership also indirectly affect MDT behavior, with effect values of 0.062 and 0.171, respectively, and confidence intervals not including 0, indicating significant indirect effects. Thus, both the direct and indirect effects of hospital culture and team leadership on MDT behavior were significant, indicating partial mediation, which supported H4 (Table 6).

## Discussion

This study used the IPO model and developed a framework to explore the influencing pathways of MDT behavior among healthcare professionals. It systematically analyzed the effects of the culture of tertiary public hospitals, team leadership, and healthcare professionals' motivation to participate in this behavior. The results indicated that hospital culture, team leadership, and motivation for participation had a significant influence on MDT behavior. In particular, a positive hospital culture and effective team leadership enhanced healthcare professionals' motivation to participate, which in turn fostered the development of MDT behavior. Moreover, participation motivation mediated the relationships between hospital culture, team leadership, and MDT behavior. This suggests that when healthcare professionals are highly motivated to participate, the effects of hospital culture and team leadership on their teamwork behavior are amplified.

### Current status of MDT among healthcare professionals

The research findings showed an average score of 3.98±0.56 for MDT among healthcare professionals, indicating that while the implementation in tertiary public hospitals in Hangzhou was generally effective, it had potential for improvement. This score was lower than that reported by Lusi et al. [54], which could be attributed to differences in the research participants. Lusi et al. [54] focused on nursing teams in tertiary hospitals, where members had long-standing relationships and a strong understanding of each other, facilitating collaboration. In contrast, MDT is a relatively new diagnostic and treatment model that is being piloted in many hospitals, indicating that team dynamics require further development.

**Table 6. Mediation effect bootstrap analysis results (standardized coefficients).**

| Path | Effect type | S.E. | Effect size | Bias-corrected 95%CI | | | Percentile 95%CI | | |
|---|---|---|---|---|---|---|---|---|---|
| | | | | Lower | Upper | P | Lower | Upper | P |
| Hospital culture → Participation motivation → MDT behavior | Total | 0.054 | 0.242 | 0.138 | 0.352 | 0.001 | 0.133 | 0.346 | 0.001 |
| | Direct | 0.051 | 0.180 | 0.079 | 0.280 | 0.001 | 0.075 | 0.277 | 0.001 |
| | Indirect | 0.024 | 0.062 | 0.022 | 0.114 | 0.004 | 0.020 | 0.112 | 0.005 |
| Team leadership → Participation motivation → MDT behavior | Total | 0.063 | 0.564 | 0.436 | 0.680 | 0.001 | 0.444 | 0.685 | 0.001 |
| | Direct | 0.068 | 0.393 | 0.258 | 0.524 | 0.001 | 0.260 | 0.526 | 0.001 |
| | Indirect | 0.043 | 0.171 | 0.101 | 0.274 | 0.001 | 0.096 | 0.266 | 0.001 |

Furthermore, the values in this study were lower than those reported by Yutong et al. [55], who examined the teamwork abilities of nursing graduate students. Their focus on disciplines with high professional compatibility likely facilitated teamwork, as it did not involve complex and challenging medical cases. Thus, while the MDT behavior values in this study were slightly low, the promotion and implementation of MDT in China are evolving. Thus, healthcare institutions should improve collaboration among MDT members through targeted training and structural enhancements while developing relevant policies to strengthen MDT models and enhance MDT effectiveness.

## The influence of hospital culture on MDT behavior

The hierarchical linear regression results indicated that hospital culture substantially influenced MDT behavior more than demographic characteristics, team leadership, and participation motivation. This aligned with the findings of Pavithra and Westbrook [56]. A positive hospital culture fosters the coordination and communication skills of healthcare professionals, thereby enhancing MDT. DiazGranados et al. [57] found that the unique culture of hospitals shaped the interactions among team members. Moreover, a systematic review revealed that hospital culture influenced the collaborative atmosphere among MDT members, ultimately impacting team collaboration [58].

Furthermore, supportive policies and a strong cooperative culture in hospitals enhance trust among team members, leading to positive collaborative interactions [15]. Safety culture, which is an integral aspect of hospital organizational culture, is vital for ensuring patient safety. Implementing safety culture training can positively influence MDT behavior, contributing to the formation of high-performance healthcare teams [59]. Thus, health administration departments and hospital managers should enhance hospital culture, cultivate a positive and harmonious work environment, encourage communication and teamwork among healthcare professionals, and support the team's sustainable development. Additionally, hospitals should consider improving healthcare professionals' coordination and communication skills through regular training and cultural development initiatives.

## The influence of team leadership on MDT behavior

This study revealed a positive relationship between team leadership and collaborative behaviors in MDT. This finding aligns with that of Chao et al. [60], who found that MDT leaders could enhance healthcare professionals' participation by using leadership skills such as effective task allocation, quick decision-making, and fostering open discussions among team members. Furthermore, a mature leadership approach fosters a democratic environment, encourages meaningful discussions, and enhances MDT [61]. Strong team leadership typically improves the overall capabilities of the team, encourages active participation from members, and supports effective MDT. Thus, hospital administration should prioritize the development of team leadership and enhance leaders' coordination and decision-making skills through targeted initiatives. Suggested initiatives include regular leadership training focusing on team coordination, decision-making, and communication skills; implementing a feedback system to encourage self-reflection and ongoing development for MDT leaders; and enhancing leadership capabilities to advance MDT.

## The influence of participation motivation on MDT behavior

This study identified a positive relationship between healthcare professionals' motivation to participate and engagement in MDT, which aligned with previous study findings [62]. This implies the proactive involvement of team members is essential for the successful functioning of MDT. Members who are highly motivated to participate in MDT tend to adopt a patient-centered approach, which enhances their ability to communicate effectively with patients and improves their overall medical experience. According to Latham et al. [63], when employees have relevant knowledge and are open to sharing it, their motivation to engage in their work increases, thereby fostering mutual learning and collaboration and enhancing team performance. Moreover, Karaferis et al. [64] found that when doctors were provided opportunities to

engage in decision-making relevant to their roles, they became invested in their work and developed collaborative relationships with their peers. Thus, hospital administration should proactively implement reward systems, career development initiatives, and enhancements to the work environment to boost healthcare professionals' motivation to engage, thereby fostering MDT.

### The mediating role of participation motivation in the relationship between hospital culture and MDT behavior

The bootstrap analysis indicated that participation motivation partially mediated the influence of hospital culture on collaborative behavior in MDT. This aligns with previous study findings, which indicate that fostering a positive hospital culture enhances healthcare professionals' motivation to engage in their work, encourages collaboration, and ultimately improves patient outcomes [16]. Furthermore, McFadden et al. [65] showed that a hospital culture that prioritizes teamwork, respect, and fairness increased healthcare professionals' motivation to participate, leading to more frequent collaboration among teams. This study revealed that a positive and proactive culture in tertiary public hospitals was correlated with increased work motivation among healthcare professionals and more frequent MDT. Thus, hospital culture and team leadership serve as input factors that directly influence multidisciplinary collaboration while impacting it indirectly through the mediation of participation motivation. Positive participation motivation is a critical link between hospital culture and collaborative behavior in MDT. These findings support the IPO model, which asserts that team inputs influence outcomes directly and through team processes [19]. Therefore, health administration departments should prioritize developing a supportive and innovative hospital culture that fosters positivity and collaboration, thereby enhancing healthcare professionals' motivation to participate, promoting MDT, and ultimately improving the quality of healthcare services.

### The mediating role of participation motivation in the relationship between team leadership and MDT behavior

The bootstrap analysis demonstrated that participation motivation partially mediated the relationship between hospital culture and teamwork behavior in multidisciplinary settings. This aligns with previous study findings indicating that participation motivation mediates the relationship between transformational leadership and organizational learning culture [17]. Transformational leadership meets employees' psychological needs and enhances their intrinsic motivation, which in turn fosters continuous learning and encourages cooperative behavior among team members [17]. Rydenfält et al. [66] used the IPO model and found that leadership styles, as input factors, influenced cohesion and communication, which are key elements of collaboration. These factors, in turn, impacted team effectiveness and patient safety, which are the desired outcomes [66].

Furthermore, Aubé et al. [67] found that proactive participation in teams mediated the relationship between perceived shared understanding and team performance, with participation motivation serving as a mediator in this relationship. This study used the IPO model, categorizing team leadership as an input factor, participation motivation as a process factor, and healthcare professionals' MDT behavior as an output factor. The findings revealed that stronger team leadership was correlated with greater participation motivation among healthcare professionals, which in turn led to more positive MDT behavior. This highlights that the motivation of healthcare professionals to participate is crucial for the effective functioning of MDT. Thus, hospital administrators should prioritize developing the leadership skills of their managers to boost healthcare professionals' participation motivation, foster MDT, improve teamwork efficiency, and ultimately enhance the quality of healthcare services.

### Implications and limitations

This study introduced participation motivation as a mediating variable and developed a mediation model that links hospital culture, team leadership, and MDT behavior. This enriches the understanding of the factors influencing teamwork behavior in tertiary public hospitals. It clarifies the roles of hospital culture and team leadership in shaping MDT behavior through participation motivation, and offers strategies to enhance collaborative behavior among team members. These insights

provide a theoretical foundation for developing MDT in public hospitals, contributing to the establishment of a patient-centered healthcare delivery system based on effective teamwork.

This study had several limitations. First, due to the cross-sectional research design, this study could only infer correlations between variables and could not establish causal relationships. Future research should consider longitudinal or experimental designs to clarify the causal relationships between MDT behaviors and their influencing factors. Second, the study's sample was restricted to tertiary public hospitals in the economically developed city of Hangzhou, which may not reflect conditions in less economically developed regions, thus limiting the generalizability of the findings. Future studies should broaden the geographical scope by including survey samples from both eastern and western provinces and cities to enhance the representativeness of the results. Third, the details of the research variables were inadequate. Future research could break down specific dimensions of hospital culture, team leadership, and participation motivation, such as categorizing team leadership into trust, teamwork skills, conflict management, and crisis handling, to obtain deeper insights.

## Conclusions

This study demonstrated that the culture of tertiary public hospitals in Hangzhou, effective team leadership, and the motivation to participate of healthcare professionals significantly influenced MDT behavior. In particular, a positive hospital culture and strong team leadership significantly enhanced teamwork behavior in multidisciplinary settings. As the culture of the hospital and team leadership improves, an increase in teamwork behavior is anticipated. Moreover, participation motivation partially mediated the relationship between hospital culture, team leadership, and teamwork behavior.

Therefore, hospital management should prioritize fostering a positive hospital culture, enhancing team leadership, and boosting healthcare professionals' participation motivation through various incentives and rewards. By adopting comprehensive strategies, hospitals can enhance teamwork among healthcare professionals and establish patient-centered multidisciplinary diagnostic and treatment teams, ultimately improving the quality of healthcare services and advancing the development of public hospitals.

## Supporting information

**S1 Appendix. Survey questionnaire for multidisciplinary teamwork behavior (English version).**
(DOCX)

**S2 Checklist. The STROBE checklist with page references.**
(DOC)

## Acknowledgments

We would like to express our deepest gratitude to the Health Commission of Zhejiang Province for their support in conducting this study. We would also like to thank all study participants for their time. Moreover, we are grateful to Jixiang Lai, who helped improve this paper.

## Author contributions

**Conceptualization:** Xianhong Huang.

**Data curation:** Zhengnan Meng.

**Formal analysis:** Jie Yang, Simin Zhu.

**Investigation:** Huaineng Wu.

**Methodology:** Jiamin Tang, Simin Zhu.

**Software:** Zhengnan Meng, Jie Yang.

**Supervision:** Xiaohe Wang.

**Validation:** Jiamin Tang, Xiaohe Wang.

**Visualization:** Zhengnan Meng.

**Writing – original draft:** Zhengnan Meng, Xianhong Huang.

**Writing – review & editing:** Zhengnan Meng, Linlin Cui.

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
