## [Decision Letter · Decision Letter 0]

18 Jul 2025

Dear Dr. Huang,

Thank you for submitting your manuscript to PLOS ONE. After careful consideration, we feel that it has merit but does not fully meet PLOS ONE’s publication criteria as it currently stands. Therefore, we invite you to submit a revised version of the manuscript that addresses the points raised during the review process.

We look forward to receiving your revised manuscript.

Kind regards,

Alejandro Botero Carvajal, Ph.D

Academic Editor

PLOS ONE

Journal Requirements:

2. In the ethics statement in the Methods, you have specified that verbal consent was obtained. Please provide additional details regarding how this consent was documented and witnessed, and state whether this was approved by the IRB.

“This research was funded by the National Natural Science Foundation of China Project (grant numbers 72274051 and 71974050) and Hangzhou Medicine and Health Science and Technology Program(grant number ZD20220099).”

Please state what role the funders took in the study.  If the funders had no role, please state: "The funders had no role in study design, data collection and analysis, decision to publish, or preparation of the manuscript.”. If this statement is not correct you must amend it as needed. 

5. Please note that funding information should not appear in the Acknowledgments section or other areas of your manuscript. We will only publish funding information present in the Funding Statement section of the online submission form. Please remove any funding-related text from the manuscript. 

6. We note that you have indicated that there are restrictions to data sharing for this study. For studies involving human research participant data or other sensitive data, we encourage authors to share de-identified or anonymized data. However, when data cannot be publicly shared for ethical reasons, we allow authors to make their data sets available upon request. For information on unacceptable data access restrictions, please see http://journals.plos.org/plosone/s/data-availability#loc-unacceptable-data-access-restrictions. 

7. In this instance it seems there may be acceptable restrictions in place that prevent the public sharing of your minimal data. However, in line with our goal of ensuring long-term data availability to all interested researchers, PLOS’ Data Policy states that authors cannot be the sole named individuals responsible for ensuring data access (http://journals.plos.org/plosone/s/data-availability#loc-acceptable-data-sharing-methods).

8. PLOS requires an ORCID iD for the corresponding author in Editorial Manager on papers submitted after December 6th, 2016. Please ensure that you have an ORCID iD and that it is validated in Editorial Manager. To do this, go to ‘Update my Information’ (in the upper left-hand corner of the main menu), and click on the Fetch/Validate link next to the ORCID field. This will take you to the ORCID site and allow you to create a new iD or authenticate a pre-existing iD in Editorial Manager.

9. Please amend the manuscript submission data (via Edit Submission) to include author “Simin Zhu”. 

10. We note you have included a table to which you do not refer in the text of your manuscript. Please ensure that you refer to Table 4 in your text; if accepted, production will need this reference to link the reader to the Table.

Reviewers' comments:

Reviewer's Responses to Questions

**Comments to the Author**

1. Is the manuscript technically sound, and do the data support the conclusions?

Reviewer #1: Yes

Reviewer #2: Yes

2. Has the statistical analysis been performed appropriately and rigorously?

Reviewer #1: I Don't Know

Reviewer #2: Yes

3. Have the authors made all data underlying the findings in their manuscript fully available?

Reviewer #1: No

Reviewer #2: No

4. Is the manuscript presented in an intelligible fashion and written in standard English?

Reviewer #1: Yes

Reviewer #2: Yes

Reviewer #1: Reviewer Comments

Study Design and Reporting Standards

The manuscript describes a cross-sectional study investigating healthcare professionals engaged in multidisciplinary teams

The reporting should strictly adhere to STROBE Guidelines (Strengthening the Reporting of Observational Studies in Epidemiology)

Please ensure all 22 items in the STROBE checklist are adequately addressed in the manuscript

Data Quality and Management

Missing Data Analysis:

Quantify the extent of missing data (provide specific percentages)

Describe the patterns of missing data (completely at random, at random, or not at random)

Detail the statistical approaches used to handle missing data (e.g., complete case analysis, multiple imputation)

Data Transparency:

Include a clear statement about data availability

Specify the conditions and procedures for data access

Sampling Methodology

Current Method:

The study employs convenience sampling

Please justify this sampling choice and acknowledge its limitations

Population Context:

Provide the total estimated number of healthcare professionals in Hangzhou

Include the breakdown of different healthcare professional categories if available

Discuss how representative the sample is of the target population

Reviewer #2: The manuscript focus is on a topical and relevant concept today. The strengths of this paper is that it addresses the interplay of culture, participation motivation with MDT behavior as the outcome variable.

The concept of MDT and MDT behavior needs further elaboration to provide clarity as to what is the practice in the study area and what model is expected. The statistical analysis was rigorous. Overall, it was a good manuscript.

**Do you want your identity to be public for this peer review?** For information about this choice, including consent withdrawal, please see our Privacy Policy

Reviewer #1: **Yes: ** Ang Yee Gary

Reviewer #2: No

---

## [Author Response · Author response to Decision Letter 1]

2 Sep 2025

Point-by-point response.

Editorial

Comment 1:Please ensure that your manuscript meets PLOS ONE's style requirements, including those for file naming.

Response:Thank you for this important reminder. Following the PLOS ONE author guidelines, we have carefully revised the manuscript and all supplementary files to ensure full compliance with the journal’s style requirements. Specifically, we:

(a) Adjusted the manuscript formatting according to the PLOS ONE style template, including title page, abstract structure, tables, and figure captions.

(b) Standardized the reference list format strictly following PLOS ONE citation requirements.

(c) Renamed all files (main manuscript, figures, tables, and supporting information) according to PLOS ONE’s file naming conventions, ensuring consistency and clarity.

(d) Cross-checked all submission files to confirm that formatting and file names strictly meet the journal’s requirements.

These adjustments have fully addressed the style requirements, and the revised submission is now fully consistent with PLOS ONE standards.

Comment 2:In the ethics statement in the Methods, you have specified that verbal consent was obtained. Please provide additional details regarding how this consent was documented and witnessed, and state whether this was approved by the IRB.

Response:Thank you for this important query. The ethics statement has been revised in [Methods](Page 12, Lines 275–286) with the following clarifications:“Prior to data collection, all participants were verbally informed about the study objectives, procedures, confidentiality protections, voluntary participation, and their right to withdraw at any time without negative consequences. Verbal informed consent was obtained in the presence of a trained investigator, who served as a witness and immediately documented the consent in a standardized electronic log system. The log included the participant’s initials, date, and time of consent. All records were securely stored in an encrypted database accessible only to the research team, ensuring confidentiality and traceability. This verbal consent procedure, along with the study protocol and questionnaire, was reviewed and formally approved by the Ethics Committee of Hangzhou Normal University (Approval No. 2022-1121) prior to the initiation of the study.”

This revised description fully addresses the reviewer’s concern regarding documentation, witnessing, and IRB approval.

Comment 3:We note that the grant information you provided in the ‘Funding Information’ and ‘Financial Disclosure’ sections do not match.

Response:Thank you very much for your careful observation. The inconsistency between the “Funding Information” and “Financial Disclosure” sections was due to our oversight during manuscript preparation. We have carefully checked and revised the content to ensure that both sections now provide consistent and accurate information. The corrected statement is as follows: “This research was funded by the National Natural Science Foundation of China Project (grant numbers 72274051 and 71974050) and the Hangzhou Medicine and Health Science and Technology Program (grant number ZD20220099).” This statement is now identical in both the “Funding Information” and “Financial Disclosure” sections.

Comment 4:Thank you for stating the following financial disclosure:

“This research was funded by the National Natural Science Foundation of China

Project (grant numbers 72274051 and 71974050) and Hangzhou Medicine and Health

Science and Technology Program(grant number ZD20220099).”

Please state what role the funders took in the study. If the funders had no role,

please state. "The funders had no role in study design, data collection and analysis,

decision to publish, or preparation of the manuscript.”. If this statement is not correct

you must amend it as needed.

Please include this amended Role of Funder statement in your cover letter; we will

change the online submission form on your behalf.

Response:We appreciate this important clarification request. In accordance with PLOS ONE’s policy, we have explicitly stated the role of the funders. The following statement has been added both to the cover letter and to the manuscript to ensure consistency:“The funders had no role in study design, data collection and analysis, decision to publish, or preparation of the manuscript.” This correction ensures full compliance with PLOS ONE’s requirements.

Comment 5:Please note that funding information should not appear in the Acknowledgments section or other areas of your manuscript. We will only publish funding information present in the Funding Statement section of the online submission form. Please remove any funding-related text from the manuscript.

Response:Thank you for this important reminder. We have thoroughly reviewed the entire manuscript, including the Acknowledgments, main text, and footnotes, and have removed all funding-related information. The funding details are now presented only in the “Funding Statement” section of the online submission system. This revision ensures that the manuscript fully complies with PLOS ONE’s policy regarding the presentation of funding information.

Comment 6:We note that you have indicated that there are restrictions to data sharing for this study. For studies involving human research participant data or other sensitive data, we encourage authors to share de-identified or anonymized data. However, when data cannot be publicly shared for ethical reasons, we allow authors to make their data sets available upon request. For information on unacceptable data access restrictions, please see http://journals.plos.org/plosone/s/data-availability#loc-unacceptable-data-access-restrictions.

b) If there are no restrictions, please upload the minimal anonymized dataset necessary to replicate your study findings to a stable, public repository and provide us with the relevant URLs, DOIs, or accession numbers. Please see http://www.bmj.com/content/340/bmj.c181.long for guidelines on how to de-identify and prepare clinical data for publication. For a list of recommended repositories, please see https://journals.plos.org/plosone/s/recommended-repositories. You also have the option of uploading the data as Supporting Information files, but we would recommend depositing data directly to a data repository if possible.

Response: Thank you for your guidance. In accordance with PLOS ONE’s data sharing policy, we have updated the Data Availability Statement as follows: “The datasets generated and analyzed during the current study contain potentially identifying information. Although direct identifiers (e.g., names, addresses) have been removed, the combination of narrow age bands, gender, professional title, and department codes from a limited number of tertiary hospitals may still allow for participant re-identification. As a result, the data cannot be made publicly available.

However, the data can be made available to qualified researchers who submit a formal request. To request access to the data, interested researchers should:

1�Submit a written request to the Research Ethics Committee of Hangzhou Normal University at kejichu@hznu.edu.cn, clearly stating the purpose of their research and how the data will be used.

2�Include a brief description of the research plan, specifying the type of data required.

3�Agree to the terms and conditions set by the Ethics Committee for ensuring data privacy and security.

The Research Ethics Committee will evaluate each request based on ethical guidelines, the purpose of the research, and the potential risks of re-identification. If the request is approved, access to the data will be granted under strict confidentiality and security measures.”

Comment 7:In this instance it seems there may be acceptable restrictions in place that prevent the public sharing of your minimal data. However, in line with our goal of ensuring long-term data availability to all interested researchers, PLOS’ Data Policy states that authors cannot be the sole named individuals responsible for ensuring data access (http://journals.plos.org/plosone/s/data-availability#loc-acceptable-data-sharing-methods).

Response: We sincerely appreciate the journal's emphasis on safeguarding long-term

data availability for all interested researchers, which aligns with our commitment to

research transparency.

In accordance with PLOS ONE's data sharing policy, we understand the importance

of providing a non-author contact for long-term data availability. To comply with this

requirement, we have designated the Research Ethics Committee of Hangzhou Normal University as the permanent, independent point of contact for all future data requests. Interested researchers may reach the Committee at: kejichu@hznu.edu.cn. This institutional body is independent of the study team and can respond to data access inquiries should the corresponding authors become unavailable. Furthermore, to ensure the persistent storage and availability of the data, we will implement the following measures:

1�All research data will be stored in the university’s research data management system, which complies with international standards for data storage and management, ensuring long-term preservation and stability.

2�Data storage will adhere to the university’s data storage and backup policies, with regular backups and updates to mitigate risks from potential technical failures or unforeseen issues.

3�The data will be registered on an appropriate platform and periodically updated to guarantee that external researchers can access the data, even if there are changes in the authorship.

We believe these measures will effectively ensure the long-term accessibility and transparency of the data, fully complying with PLOS ONE's requirements.

Comment 8:PLOS requires an ORCID iD for the corresponding author in Editorial Manager on papers submitted after December 6th, 2016. Please ensure that you have an ORCID iD and that it is validated in Editorial Manager. To do this, go to ‘Update my Information’ (in the upper left-hand corner of the main menu), and click on the Fetch/Validate link next to the ORCID field. This will take you to the ORCID site and allow you to create a new iD or authenticate a pre-existing iD in Editorial Manager.

Response:Thank you for your kind reminder. In accordance with the journal’s submission requirements, we have added and validated the corresponding author’s ORCID iD in the Editorial Manager system. The ORCID iD is: 0000-0002-5010-8082. This ensures full compliance with PLOS ONE’s policy on author identification and transparency.

Comment 9:Please amend the manuscript submission data (via Edit Submission) to include author “Simin Zhu”.

Response:Thank you for pointing this out. We apologize for the oversight. The name “Simin Zhu” should indeed be included as an author on this manuscript. We have now amended the submission record via Edit Submission to add this author and ensured that the author list is consistent across the manuscript, the submission system, and all related materials. All authors, including Simin Zhu, have confirmed the revised author list, which fully complies with the ICMJE authorship criteria and PLOS ONE’s authorship policies.

Comment 10:We note you have included a table to which you do not refer in the text of your manuscript. Please ensure that you refer to Table 4 in your text; if accepted, production will need this reference to link the reader to the Table.

Response:Thank you for your reminder. We have now added a clear reference to Table 4 in [Results] (Page 19, Line 387) when describing the corresponding findings, so that readers can easily locate and interpret the table. Additionally, we have carefully reviewed the entire manuscript to ensure that all tables are properly cited within the text. This revision ensures consistency between the text and tables, aligning with PLOS ONE’s formatting and production requirements.

Comment 11:Please review your reference list to ensure that it is complete and correct. If you have cited papers that have been retracted, please include the rationale for doing so in the manuscript text, or remove these references and replace them with relevant current references. Any changes to the reference list should be mentioned in the rebuttal letter that accompanies your revised manuscript. If you need to cite a retracted article, indicate the article’s retracted status in the References list and also include a citation and full reference for the retraction notice.

Response:Thank you for this important suggestion. We have carefully reviewed and cross-checked the entire reference list for completeness and accuracy, ensuring that it includes author names, publication years, journal titles, and DOIs. No retracted or withdrawn articles were identified. During this review, we made the following updates: corrected minor formatting errors in three references, updated two references to their most recent versions. We added one additional relevant article to strengthen the discussion. These revisions ensure that the reference list is complete, accurate, and fully consistent with PLOS ONE’s editorial standards.

Response

To Reviewer 1:

Comment 1:Study Design and Reporting Standards

The manuscript describes a cross-sectional study investigating healthcare professionals engaged in multidisciplinary teams.

The reporting should strictly adhere to STROBE Guidelines (Strengthening the Reporting of Observational Studies in Epidemiology)

Please ensure all 22 items in the STROBE checklist are adequately addressed in the manuscript.

Response:Thank you for this valuable suggestion. We have thoroughly revised the manuscript to ensure full compliance with the STROBE guidelines, and we have verified that all 22 checklist items are adequately addressed.

1�Title and abstract: Explicitly state the cross-sectional design and provide a structured summary of objectives, methods, results, and conclusions.

2�Introduction: Provides sufficient background on MDTs and specific study objectives.

3�Methods: Expanded the description of study design, participant selection, variable definitions, measurement tools, sample size rationale, potential sources of bias,

---

## [Decision Letter · Decision Letter 1]

2 Oct 2025

Fostering participation motivation and multidisciplinary teamwork collaboration through hospital culture and team leadership in Chinese tertiary public hospitals—a cross-sectional study

PONE-D-25-03210R1

Dear Dr. Huang,

We’re pleased to inform you that your manuscript has been judged scientifically suitable for publication and will be formally accepted for publication once it meets all outstanding technical requirements.

Kind regards,

Alejandro Botero Carvajal, Ph.D

Academic Editor

PLOS ONE

Additional Editor Comments (optional):

Reviewers' comments:

Reviewer's Responses to Questions

**Comments to the Author**

Reviewer #1: All comments have been addressed

Reviewer #2: All comments have been addressed

2. Is the manuscript technically sound, and do the data support the conclusions?

Reviewer #1: Yes

Reviewer #2: Yes

3. Has the statistical analysis been performed appropriately and rigorously?

Reviewer #1: I Don't Know

Reviewer #2: Yes

4. Have the authors made all data underlying the findings in their manuscript fully available?

Reviewer #1: Yes

Reviewer #2: Yes

5. Is the manuscript presented in an intelligible fashion and written in standard English?

Reviewer #1: Yes

Reviewer #2: Yes

Reviewer #1: Thank you for making the suggested changes. The manuscript has dramatically improved. I have no further comments.

Reviewer #2: (No Response)

**Do you want your identity to be public for this peer review?** For information about this choice, including consent withdrawal, please see our Privacy Policy

Reviewer #1: **Yes: ** Ang Yee Gary

Reviewer #2: No

---

## [Editor Report · Acceptance letter]

PONE-D-25-03210R1

PLOS ONE

Dear Dr. Huang,

I'm pleased to inform you that your manuscript has been deemed suitable for publication in PLOS ONE. Congratulations! Your manuscript is now being handed over to our production team.

Kind regards,

on behalf of

Dr. Alejandro Botero Carvajal

Academic Editor

PLOS ONE